# Convergence of Collocation Methods for One Class of Impulsive Delay Differential Equations

**Zhiwei Wang, Guilai Zhang \* and Yang Sun**

College of Sciences, Northeastern University, Shenyang 110819, China; 2172045@stu.neu.edu.cn (Z.W.); 2101906@stu.neu.edu.cn (Y.S.)

\* Correspondence: zhangguilai@neuq.edu.cn

**Abstract:** This paper is concerned with collocation methods for one class of impulsive delay differential equations (IDDEs). Some results for the convergence, global superconvergence and local superconvergence of collocation methods are given. We choose a suitable piecewise continuous collocation space to obtain high-order numerical methods. Some illustrative examples are given to verify the theoretical results.

**Keywords:** impulsive delay differential equations; collocation methods; convergence; superconvergence

**MSC:** 65L03





## 1. Introduction

Impulsive differential equations appear to represent models of several real-life phenomena. In recent decades, systems with impulse effects have arisen in control theory, medicine, biotechnology, economics, population growth, etc. Some work on these systems was presented [1–5]. In recent years, there has been increasing attention on the initial value problem of IDDEs. The corresponding theory of the exact solutions of IDDEs has been studied from different angles (see [6–12]): oscillation, stability, asymptotic stability and exponential stability in some specific classes of IDDEs.

Collocation methods as numerical methods have a wide range of applications in the treatment of integral–algebraic equations [13–16], Volterra integral equations [17–19] and delay differential equations [20–22]. Specifically, the convergence of the collocation methods has received a lot of attention, such as the convergence of collocation methods for weakly singular Volterra integral equations [23], the superconvergence of collocation methods for first-kind Volterra integral equations [24], the convergence of collocation methods for Volterra integral equations [25], the convergence of multistep collocation methods for integral–algebraic equations [16], etc. But to the best of our knowledge, there are no articles referring to the convergence of the collocation method for IDDEs.

In this paper, we consider the following impulsive delay differential equation with collocation methods:

$$\begin{cases} y'(t) = p(t)y(t) + q(t)y(t - \tau), & t \neq k\tau, k = 1, 2, \cdots, t \in I, \\ \triangle y = B_k y, & t = k\tau, k = 1, 2, \cdots, \\ y(t) = \phi(t), & t \in [-\tau, 0], \end{cases} \quad (1)$$

where $I := [0, T]$, $\triangle y = y(t^+) - y(t)$, $y(t^+)$ is the right limit of $y(t)$, $p : I \to \mathbb{R}, q : I \to \mathbb{R}$ are two given functions and sufficiently smooth, $\tau > 0$ is a positive constant, $\phi$ is a continuous function on $[-\tau, 0]$ and $y'(t)$ denotes the left-hand derivative of $y(t)$.

The rest of the present paper is organized as follows: Firstly, the existence and uniqueness of collocation methods are presented in Section 2. In Section 3, the global convergence of collocation methods is analytically derived. Following that, Section 4 gives the global

and local superconvergence of properties. Finally, two numerical experiments are given in Section 5.

**Definition 1** (Jurang Yan [8]). *The function $y : I \to R$ is said to be a solution of system (1) when the following conditions are satisfied:*

1. $y(t) = \phi(t), t \in [-\tau, 0]$;
2. *for $t \in I, t \neq k\tau$, the function $y(t)$ is differentiable and $y'(t) = p(t)y(t) + q(t)y(t - \tau)$;*
3. *the function $y(t)$ is left-continuous in $I$, and if $t \in I$ and $t = k\tau$, then $y(t^+) = (1 + B_k)y(t)$, $y(t^-) = y(t)$;*
4. $B_k \in (-\infty, -1) \cup (-1, +\infty)$ *are constants, $k = 1, 2, \cdots$.*

## 2. Collocation Methods

For ease of notation, we assume that $T = N\tau$, $N$ is a positive integer. All $k\tau$, $k = 1, 2, \cdots, N$, are chosen as numerical nodes to ensure the convergence of collocation methods. Define a positive integer $p \geqslant 1$ and the stepsize $h = \frac{\tau}{p}$ corresponding to the given intervals $(t_n, t_{n+1})$. $t_n = nh$ are fixed time. The global mesh $I_h$ on $I$ is defined by

$$I_h := \{t_n : 0 = t_0 < t_1 < \cdots < t_{Np} = T\}.$$

Firstly, we will choose the collocation points as follows:

$$X_h := \{t_{n,i} = t_n + c_i h : 0 < c_1 < \cdots < c_m \leqslant 1\},$$

where $\{c_i\}$ indicates a series of collocation parameters. Define $\sigma_n := (t_n, t_{n+1}]$. The exact solution can be approximated by a collocation solution in the piecewise polynomial space

$$\widetilde{S_m^{(0)}}(I_h) := \left\{ v : v|_{\sigma_n} \in \pi_m, \begin{cases} \triangle v = 0, & \text{if } t \neq k\tau, t \in I \\ \triangle v = B_k v, & t = k\tau \end{cases} \right\},$$

where $\pi_m$ denotes the space of all real polynomials of degree not exceeding $m$ (see [17,21]), and $\triangle v = v(t^+) - v(t)$. The collocation solution $u_h$ is the element of the piecewise polynomial space that satisfies the following equation:

$$\begin{cases} u_h'(t) = p(t)u_h(t) + q(t)u_h(t - \tau), & t \neq k\tau, t \in X_h, \\ \triangle u_h\left(t_{kp}\right) = B_k u_h\left(t_{kp}\right), & k = 1, 2, \cdots, \\ u_h(t) = \phi(t), & t \in [-\tau, 0], \end{cases} \quad (2)$$

where $u_h(t)$ and $u_h'(t)$ are left-continuous.

Setting $Y_{n,j} := u_h'(t_n + c_j h)$, we have

$$u_h'(t_n + vh) = \sum_{j=1}^m L_j(v)Y_{n,j}, \quad (3)$$

where $L_j(v)$ denotes the following Lagrange fundamental corresponding to the collocation parameters $\{c_i\}$ (see [17,21]):

$$L_j(v) = \prod_{i=1,i\neq j}^m \frac{v - c_i}{c_j - c_i}.$$

Integrating (3), we can obtain

$$u_h(t_n + vh) = u_h\left(t_n^+\right) + h \sum_{j=1}^m \beta_j(v)Y_{n,j}, v \in (0,1], \quad (4)$$

where

$$\beta_j(v) = \int_0^v L_j(s)ds.$$

According to the definition of $\widetilde{S_m^{(0)}}(I_h)$, we have

$$u_h(t_n^+) = \begin{cases} u_h(t_n), & t_n \neq k\tau, k = 1, 2, \cdots, \\ (1 + B_k)u_h(t_n), & t_n = k\tau. \end{cases} \tag{5}$$

By (2) and (4), we obtain

$$\begin{aligned} Y_{n,i} &= p(t_{n,i})u_h(t_{n,i}) + q(t_{n,i})u_h(t_{n,i} - \tau) \\ &= p(t_{n,i})\left[u_h(t_n^+) + h\sum_{j=1}^m \beta_j(v)Y_{n,j}\right] \\ &\quad + q(t_{n,i})\left[u_h\left(t_{n-p}^+\right) + h\sum_{j=1}^m \beta_j(v)Y_{n-p,j}\right], \end{aligned} \tag{6}$$

where $a_{ij} := \beta_j(c_i)$. Let

$$\tilde{P}_n := \text{diag}(p(t_{n,i})), A = (a_{ij}) \in L(\mathbb{R}^m), P_n := \tilde{P}_n A,$$

$$\tilde{Q}_n := \text{diag}(q(t_{n,i})), A = (a_{ij}) \in L(\mathbb{R}^m), Q_n := \tilde{Q}_n A,$$

$$\beta(v) := (\beta_1(v), \beta_2(v), \cdots, \beta_m(v))^T, Y_n := (Y_{n,1}, Y_{n,2}, \cdots, Y_{n,m})^T, e := \left(\underbrace{1, \cdots, 1}_{m}\right)^T.$$

Then

$$[I_{m \times m} - hP_n]Y_n = \left[\tilde{P}_n u_h(t_n^+) + \tilde{Q}_n u_h\left(t_{n-p}^+\right)\right]e + hQ_n Y_{n-p}. \tag{7}$$

When the solution $Y_n$ has been found by (6), the collocation solution on the interval $(t_n, t_{n+1}]$ is determined by

$$u_h[(t_n + vh)] = u_h(t_n^+) + h\beta^T(v)Y_n, v \in (0, 1]. \tag{8}$$

**According to [17], the following theorem is given without proof.**

**Theorem 1.** *There exists an $\bar{h} > 0$ such that for the mesh diameter $h$ belonging to the interval $(0, \bar{h})$, (7) has unique solutions $Y_n \in \mathbb{R}^m$. Then, the collocation solution $u_h \in \widetilde{S_m^{(0)}}(I_h)$ for impulsive delay differential Equation (1) is unique and is given by (8) on the subinterval $(t_n, t_{n+1}]$.*

## 3. Global Convergence

In the following section, the global convergence of the collocation solution for IDDEs will be analyzed.

**Theorem 2.** *If $p, q \in C^m(I)$ and the collocation solution $u_h$ for (1) is defined by (2), then there exists two constants $C_0$ and $C_1$ which are independent of $h$, satisfying*

$$\|y - u_h\|_\infty := \max_{t \in I}|y(t) - u_h(t)| \leqslant C_0\left\|y^{(m+1)}\right\|_\infty h^m, \tag{9}$$

$$\|y' - u_h'\|_\infty := \sup_{t \in I}|y'(t) - u_h'(t)| \leqslant C_1\left\|y^{(m+1)}\right\|_\infty h^m, \tag{10}$$

*for $h \in (0, \bar{h})$ and any collocation parameters with $0 < c_1 < \cdots < c_m \leqslant 1$.*

**Proof.** Assume that $p, q \in C^m(I)$ implies $y \in C^{m+1}(\sigma_n)$ and $y' \in C^m(\sigma_n)$. The collocation error $e_h(t) := y(t) - u_h(t)$ satisfies the equation

$$e'_h(t) = p(t)e_h(t) + q(t)e_h(t - \tau), \quad t \neq k\tau, \quad t \in X_h, \tag{11}$$

with $e_h(t) = 0, t \leqslant 0$. By Peano's theorem [17], we can obtain that

$$y'(t_n + vh) = \sum_{j=1}^{m} L_j(v)Z_{n,j} + h^m R^{(1)}_{m+1,n}(v), \quad v \in (0, 1], \tag{12}$$

where

$$R^{(1)}_{m+1,n}(v) := \int_0^1 K_m(v, z)y^{(m+1)}(t_n + zh)dz,$$

$$K_m(v, z) := \frac{1}{(m-1)!}\left\{(v - z)_+^{m-1} - \sum_{k=1}^{m} L_k(v)(c_k - z)_+^{m-1}\right\}, v \in (0, 1],$$

and $Z_{n,j} := y'(t_{n,j})$. Integrating (12), we have

$$y(t_n + vh) = y(t_n^+) + h\sum_{j=1}^{m} \beta_j(v)Z_{n,j} + h^{m+1}R_{m+1,n}(v), v \in (0, 1], \tag{13}$$

where

$$R_{m+1,n}(v) := \int_0^v R^{(1)}_{m+1,n}(v)dv,$$

and

$$y(t_n^+) = \begin{cases} y(t_n), & t_n \neq k\tau, k = 1, 2, \cdots, \\ (1 + B_k)y(t_n), & t_n = k\tau. \end{cases}$$

Let $\varepsilon_{n,j} := Z_{n,j} - Y_{n,j}$. Comparing (4) and (13), we obtain

$$e_h(t_n + vh) = e_h(t_n^+) + h\sum_{j=1}^{m} \beta_j(v)\varepsilon_{n,j} + h^{m+1}R_{m+1,n}(v), v \in (0, 1], \tag{14}$$

where

$$e_h(t_n^+) = \begin{cases} e_h(t_n), & t_n \neq k\tau, k = 1, 2, \cdots, \\ (1 + B_k)e_h(t_n), & t_n = k\tau. \end{cases} \tag{15}$$

Due to (3) and (12), we can obtain that

$$e'_h(t_n + vh) = \sum_{j=1}^{m} L_j(v)\varepsilon_{n,j} + h^m R^{(1)}_{m+1,n}(v), v \in (0, 1]. \tag{16}$$

By the definition of $\varepsilon_{n,j}$ and (14), we obtain

$$
\begin{aligned}
\varepsilon_{n,i} &= y'(t_{n,i}) - u'_h(t_{n,i}) \\
&= p(t_{n,i})e_h(t_n + vh) + q(t_{n,i})e_h[(t_n + vh) - \tau] \\
&= p(t_{n,i})e_h(t_n + vh) + q(t_{n,i})e_h\big[(t_{n-p} + vh)\big] \\
&= p(t_{n,i})\Bigg[e_h(t_n^+) + h\sum_{j=1}^{m} a_{ij}\varepsilon_{n,j} + h^{m+1}R_{m+1,n}(c_i)\Bigg] \\
&\quad + q(t_{n,i})\Bigg[e_h\Big(t_{n-p}^+\Big) + h\sum_{j=1}^{m} a_{ij}\varepsilon_{n-p,j} + h^{m+1}R_{m+1,n-p}(c_i)\Bigg],
\end{aligned}
\tag{17}
$$

i.e.,

$$
\begin{aligned}
[I_{m\times m} - hP_n]\varepsilon_n &= \Big[\widetilde{P}_n e_h(t_n^+) + \widetilde{Q}_n e_h\Big(t_{n-p}^+\Big)\Big]e + h^{m+1}\widetilde{P}_n R_{m+1,n} \\
&\quad + hQ_n\varepsilon_{n-p} + h^{m+1}\widetilde{Q}_n R_{m+1,n-p},
\end{aligned}
\tag{18}
$$

where $R_{m+1,n} := (R_{m+1,n}(c_1), \ldots, R_{m+1,n}(c_m))^T$ and $\varepsilon_n := (\varepsilon_{n,1}, \varepsilon_{n,2}, \ldots, \varepsilon_{n,m})^T$. For ease of notation, we assume $n = pk + l\,(l = 1, 2, \ldots, p)$, then $t_n = t_{pk+l} \in (k\tau, (k+1)\tau]$. By (14) and (15),

$$
\begin{aligned}
e_h(t_n^+) &= e_h\Big(t_{pk+l}^+\Big) = W_{k+1}e_h\Big(t_{pk+l}\Big) = W_{k+1}e_h\Big(t_{pk+l-1} + h\Big) \\
&= W_{k+1}\Bigg[e_h\Big(t_{pk+l-1}\Big) + h\sum_{j=1}^{m} b_j\varepsilon_{pk+l-1,j} + h^{m+1}R_{m+1,pk+l-1}(1)\Bigg] \\
&= \cdots = W_{k+1}\Bigg[e_h\Big(t_{pk}^+\Big) + \sum_{i=pk}^{n-1} h\sum_{j=1}^{m} b_j\varepsilon_{i,j} + \sum_{i=pk}^{n-1} h^{m+1}R_{m+1,i}(1)\Bigg] \\
&= W_{k+1}\Bigg[(1 + B_k)e_h\Big(t_{pk}\Big) + \sum_{i=pk}^{n-1} h\sum_{j=1}^{m} b_j\varepsilon_{i,j} + \sum_{i=pk}^{n-1} h^{m+1}R_{m+1,i}(1)\Bigg] \\
&= \cdots = W_{k+1}\prod_{d=1}^{k}(1 + B_d)\Bigg[\sum_{i=0}^{p-1} h\sum_{j=1}^{m} b_j\varepsilon_{i,j} + \sum_{i=0}^{p-1} h^{m+1}R_{m+1,i}(1)\Bigg] \\
&\quad + W_{k+1}\prod_{d=2}^{k}(1 + B_d)\Bigg[\sum_{i=p}^{2p-1} h\sum_{j=1}^{m} b_j\varepsilon_{i,j} + \sum_{i=p}^{2p-1} h^{m+1}R_{m+1,i}(1)\Bigg] \\
&\quad + \cdots + W_{k+1}(1 + B_k)\Bigg[\sum_{i=(k-1)p}^{kp-1} h\sum_{j=1}^{m} b_j\varepsilon_{i,j} + \sum_{i=(k-1)p}^{kp-1} h^{m+1}R_{m+1,i}(1)\Bigg] \\
&\quad + W_{k+1}\Bigg[\sum_{i=kp}^{pk+l-1} h\sum_{j=1}^{m} b_j\varepsilon_{i,j} + \sum_{i=kp}^{pk+l-1} h^{m+1}R_{m+1,i}(1)\Bigg],
\end{aligned}
$$

where $b_j := \beta_j(1), e_h(0^+) = 0$, and

$$
W_k := \begin{cases} 1 + B_k, & \text{if } l = p, \\ 1, & l \neq p. \end{cases}
\tag{19}
$$

Hence,

$$
\begin{aligned}
e_h\left(t_{n-p}^+\right) &= e_h\left(t_{p(k-1)+l}^+\right) = W_k e_h\left(t_{p(k-1)+l}\right) = W_k e_h\left(t_{p(k-1)+l-1} + h\right) \\
&= \cdots = W_k \prod_{d=1}^{k-1}(1+B_d)\left[\sum_{i=0}^{p-1} h \sum_{j=1}^{m} b_j \varepsilon_{i,j} + \sum_{i=0}^{p-1} h^{m+1} R_{m+1,i}(1)\right] \\
&\quad + W_k \prod_{d=2}^{k-1}(1+B_d)\left[\sum_{i=p}^{2p-1} h \sum_{j=1}^{m} b_j \varepsilon_{i,j} + \sum_{i=p}^{2p-1} h^{m+1} R_{m+1,i}(1)\right] \\
&\quad + \cdots + W_k(1+B_{k-1})\left[\sum_{i=(k-2)p}^{kp-p-1} h \sum_{j=1}^{m} b_j \varepsilon_{i,j} + \sum_{i=(k-2)p}^{kp-p-1} h^{m+1} R_{m+1,i}(1)\right] \\
&\quad + W_k\left[\sum_{i=kp-p}^{pk-p+l-1} h \sum_{j=1}^{m} b_j \varepsilon_{i,j} + \sum_{i=kp-p}^{pk-p+l-1} h^{m+1} R_{m+1,i}(1)\right],
\end{aligned}
\tag{20}
$$

where $b := (b_1, b_2, \cdots, b_m)^T$. In view of Theorem 1, we can easily obtain that the matrices $(I_m - hP_n - hQ_n)$ have bounded inverses whenever $h \in (0, \bar{h})$, and there exists a constant $D_0 < \infty$ such that

$$
\left\|(I_m - hP_n - hQ_n)^{-1}\right\|_1 \leqslant D_0, n = 0, 1, 2, \cdots.
$$

By (18),

$$
\begin{aligned}
\|\varepsilon_n\|_1 \leqslant D_0 \| &h^{m+1}\widetilde{P}_n R_{m+1,n} + h^{m+1}\widetilde{Q}_n R_{m+1,n-p} \\
&+ \tilde{P}_n e W_{k+1} \prod_{d=1}^{k}(1+B_d)\left[\sum_{i=0}^{p-1} h \sum_{j=1}^{m} b_j \varepsilon_{i,j} + \sum_{i=0}^{p-1} h^{m+1} R_{m+1,i}(1)\right] \\
&+ \tilde{P}_n e W_{k+1} \prod_{d=2}^{k}(1+B_d)\left[\sum_{i=p}^{2p-1} h \sum_{j=1}^{m} b_j \varepsilon_{i,j} + \sum_{i=p}^{2p-1} h^{m+1} R_{m+1,i}(1)\right] \\
&+ \cdots + \widetilde{P}_n e W_{k+1}(1+B_k)\left[\sum_{i=(k-1)p}^{kp-1} h \sum_{j=1}^{m} b_j \varepsilon_{i,j} + \sum_{i=(k-1)p}^{kp-1} h^{m+1} R_{m+1,i}(1)\right] \\
&+ \widetilde{P}_n e W_{k+1}\left[\sum_{i=kp}^{pk+l-1} h \sum_{j=1}^{m} b_j \varepsilon_{i,j} + \sum_{i=kp}^{pk+l-1} h^{m+1} R_{m+1,i}(1)\right] \\
&+ \tilde{Q}_n e W_k \prod_{d=1}^{k-1}(1+B_d)\left[\sum_{i=0}^{p-1} h \sum_{j=1}^{m} b_j \varepsilon_{i,j} + \sum_{i=0}^{p-1} h^{m+1} R_{m+1,i}(1)\right] \\
&+ \tilde{Q}_n e W_k \prod_{d=2}^{k-1}(1+B_d)\left[\sum_{i=p}^{2p-1} h \sum_{j=1}^{m} b_j \varepsilon_{i,j} + \sum_{i=p}^{2p-1} h^{m+1} R_{m+1,i}(1)\right] \\
&+ \cdots + \tilde{Q}_n e W_k(1+B_{k-1})\left[\sum_{i=(k-2)p}^{kp-p-1} h \sum_{j=1}^{m} b_j \varepsilon_{i,j} + \sum_{i=(k-2)p}^{kp-p-1} h^{m+1} R_{m+1,i}(1)\right] \\
&+ \widetilde{Q}_n e W_k\left[\sum_{i=kp-p}^{pk-p+l-1} h \sum_{j=1}^{m} b_j \varepsilon_{i,j} + \sum_{i=kp-p}^{pk-p+l-1} h^{m+1} R_{m+1,i}(1)\right] \|_1.
\end{aligned}
$$

Because $|B_i|(i = 1, 2, \ldots, k)$ is finite, there exists a constant $R(R > 1)$, satisfying $\left|\prod_{d=1}^{k}(1+B_d)\right| \leqslant R(d = 1, 2, \cdots, k)$. Let

$$
P_0 := \|p(t)\|_\infty, Q_0 := \|q(t)\|_\infty, M_{m+1} := \left\|y^{(m+1)}\right\|_\infty,
$$

$$K_m := \max_{v \in [0,1]} \int_0^v |K_m(v,z)|dz, \bar{b} := \max_{(j)} |b_j|.$$

Consequently, we have

$$\|\varepsilon_n\|_1 \leqslant D_0 |W_{k+1}|R| \|\tilde{P}_n e\|_1 \left[ \sum_{i=0}^{n-1} h \left| b^T \varepsilon_i \right| + \sum_{i=0}^{n-1} h^{m+1} |R_{m+1,i}(1)| \right] + D_0 h^{m+1} \left\| \widetilde{P}_n R_{m+1,n} \right\|_1$$

$$+ D_0 |W_k|R| \|\tilde{Q}_n e\| \left[ \sum_{i=0}^{n-p-1} h \left| b^T \varepsilon_i \right| + \sum_{i=0}^{n-p-1} h^{m+1} |R_{m+1,i}(1)| \right] + D_0 h^{m+1} \left\| \tilde{Q}_n R_{m+1,n-p} \right\|_1$$

$$\leqslant D_0 |W_{k+1}|R| \|\tilde{P}_n e\|_1 \left[ \sum_{i=0}^{n-1} h \left| b^T \varepsilon_i \right| + \sum_{i=0}^{n-1} h^{m+1} |R_{m+1,i}(1)| \right] + D_0 h^{m+1} \left\| \tilde{P}_n R_{m+1,n} \right\|_1$$

$$+ D_0 |W_k|R| \|\tilde{Q}_n e\|_1 \left[ \sum_{i=0}^{n-1} h \left| b^T \varepsilon_i \right| + \sum_{i=0}^{n-1} h^{m+1} |R_{m+1,i}(1)| \right] + D_0 h^{m+1} \left\| \tilde{Q}_n R_{m+1,n-p} \right\|_1$$

$$\leqslant D_0 \max\{|W_k|, |W_{k+1}|\} m(P_0 + Q_0) R \bar{b} \sum_{i=0}^{n-1} h \|\varepsilon_i\|_1$$

$$+ D_0 \max\{|W_k|, |W_{k+1}|\} m(P_0 + Q_0) R \left( \sum_{i=0}^{n-1} h \right) K_m M_{m+1} h^m$$

$$+ D_0 m(P_0 + Q_0) m K_m M_{m+1} h^{m+1}$$

$$\leqslant D_0 m(P_0 + Q_0) R^2 \bar{b} \sum_{i=0}^{n-1} h \|\varepsilon_i\|_1$$

$$+ \left( D_0 m(P_0 + Q_0) R^2 T K_m + D_0 m(P_0 + Q_0) m K_m T \right) M_{m+1} h^m$$

$$=: \gamma_0 \sum_{i=0}^{n-1} h \|\varepsilon_i\|_1 + \gamma_1 M_{m+1} h^m,$$

with obvious meaning of $\gamma_0, \gamma_1$. Due to the discrete Gronwall inequality [17], we obtain

$$\|\varepsilon_n\|_1 \leqslant \gamma_1 M_{m+1} h^m \exp(\gamma_0 T) =: B M_{m+1} h^m, n = 0, 1, \cdots,$$

and

$$\left| e_h(t_n^+) \right| \leqslant R |W_{k+1}| \bar{b} \sum_{i=0}^{n-1} h \|\varepsilon_i\|_1 + R |W_{k+1}| h^m \left( \sum_{i=0}^{n-1} h \right) K_m M_{m+1}.$$

By (14) and (16),

$$|e_h(t_n + vh)| \leqslant \left| e_h(t_n^+) \right| + h\bar{\beta} \|\varepsilon_n\|_1 + h^{m+1} K_m M_{m+1}$$

$$\leqslant R |W_{k+1}| \bar{b} \sum_{i=0}^{n-1} h \|\varepsilon_i\|_1 + R |W_{k+1}| \left( \sum_{i=0}^{n-1} h \right) K_m M_{m+1} h^m$$

$$+ h\bar{\beta} \|\varepsilon_n\|_1 + h^{m+1} K_m M_{m+1}$$

$$\leqslant \left[ R |W_{k+1}| \left( \sum_{i=0}^{n-1} h \right) \bar{b} B + R |W_{k+1}| \left( \sum_{i=0}^{n-1} h \right) K_m + h\bar{\beta} B + h K_m \right] M_{m+1} h^m$$

$$\leqslant \left[ R^2 T \bar{b} B + R^2 T K_m + T \bar{\beta} B + T K_m \right] M_{m+1} h^m =: C_0 M_{m+1} h^m,$$

and

$$\left| e_h'(t_n + vh) \right| \leqslant \Lambda B M_{m+1} h^m + h^m K_m M_{m+1}$$

$$= (\Lambda B + K_m) M_{m+1} h^m \tag{21}$$

$$=: C_1 M_{m+1} h^m,$$

where

$$\bar{\beta} := \max_{(j)} \|\beta_j\|_\infty, \Lambda := \max_{(j)} \|L_j\|_\infty.$$

The proof of Theorem 2 is complete. □

## 4. Global Superconvergence and Local Superconvergence

In this part, the global superconvergence of the collocation solution is discussed first and the local superconvergence is analyzed later.

**Theorem 3.** *Let the given function in* (1) *satisfy* $p, q \in C^d(I), \phi \in C^{d+1}[-\tau, 0]$ *,* $d \geqslant m + 1$. *Assume that the m collocation parameters* $\{c_i\}$ *are subject to the orthogonality condition*

$$J_0 := \int_0^1 \prod_{i=1}^m (s - c_i) ds = 0. \tag{22}$$

*Then, the corresponding collocation solution* $u_h$ *on I satisfies the following conditions:*

$$\|y - u_h\|_\infty \leqslant C_2 h^{m+1}, \tag{23}$$

$$\|y' - u_h'\|_\infty \leqslant C_3 h^m, \tag{24}$$

*where* $h \in (0, \bar{h})$, $C_2$ *and* $C_3$ *are two constants which are independent of h.*

**Proof.** The (24) can be obtained with (21). The following discussion is for (23). We define the defect $\delta_h(t)$ by

$$\delta_h(t) := -u_h'(t) + p(t)u_h(t) + q(t)u_h(t - \tau), t \in I. \tag{25}$$

By (1), we can easily obtain the following form:

$$\delta_h(t) := e_h'(t) - p(t)e_h(t) - q(t)e_h(t - \tau), t \in I, \tag{26}$$

and $\delta_h(t) = 0$ for all $t \in X_h$. Due to Theorem 2, we can obtain that

$$\|\delta_h\|_\infty \leqslant C_1 M_{m+1} h^m + P_0 C_0 M_{m+1} h^m + Q_0 C_0 M_{m+1} h^m =: D_1 M_{m+1} h^m, \tag{27}$$

for any $c_i$ in $\{c_i : i = 1, 2, \cdots, m, 0 < c_i \leq 1\}$.

Here, $e_h(t)$ can be treated as the solution of the following equation:

$$\begin{cases} e_h'(t) = p(t)e_h(t) + q(t)e_h(t - \tau) + \delta_h(t), & t \neq k\tau, t \in I, \\ e_h(t^+) = (1 + B_k)e_h(t), & t = k\tau, \\ e_h(t) = 0, & t \in [-\tau, 0]. \end{cases} \tag{28}$$

Let $r(t, s)$ denote the resolvent of (1)

$$r(t, s) := \exp\left(\int_s^t p(v)dv\right), r \in C^{m+1}(D),$$

where $D := \{(t, s) : 0 \leqslant s \leqslant t \leqslant T\}$. So, for $t \in (0, \tau]$, we have

$$e_h(t) = \int_0^t r(t, s)(q(s)e_h(s - \tau) + \delta_h(s))ds,$$

for $t \in (\tau, 2\tau]$, we obtain

$$e_h(t) = (1 + B_1)r(t, \tau) \int_0^\tau r(\tau, s)(q(s)e_h(s - \tau) + \delta_h(s))ds$$
$$+ \int_\tau^t r(t, s)(q(s)e_h(s - \tau) + \delta_h(s))ds,$$

for $t \in (2\tau, 3\tau]$, we can obtain that

$$e_h(t) = (1 + B_2)r(t, 2\tau)\Bigg[(1 + B_1)r(2\tau, \tau)\int_0^\tau r(\tau, s)(q(s)e_h(s - \tau) + \delta_h(s))ds$$
$$+ \int_\tau^{2\tau} r(t, s)(q(s)e_h(s - \tau) + \delta_h(s))ds\Bigg] + \int_{2\tau}^t r(t, s)(q(s)e_h(s - \tau) + \delta_h(s))ds,$$

for $t \in (k\tau, (k+1)\tau]$, $e_h(t)$ can be expressed by

$$e_h(t)$$
$$= r(t, k\tau)\prod_{d=1}^k (1 + B_d)\prod_{\mu=2}^k r(\mu\tau, (\mu - 1)\tau)\int_0^\tau r(\tau, s)(q(s)e_h(s - \tau) + \delta_h(s))ds$$
$$+ r(t, k\tau)\prod_{d=2}^k (1 + B_d)\prod_{\mu=3}^k r(\mu\tau, (\mu - 1)\tau)\int_\tau^{2\tau} r(2\tau, s)(q(s)e_h(s - \tau) + \delta_h(s))ds$$
$$+ \cdots$$
$$+ r(t, k\tau)(1 + B_k)\int_{(k-1)\tau}^{k\tau} r(k\tau, s)(q(s)e_h(s - \tau) + \delta_h(s))ds$$
$$+ \int_{k\tau}^t r(t, s)(q(s)e_h(s - \tau) + \delta_h(s))ds.$$

For ease of notation, we assume that $n = pk + l$, $(l = 1, 2, \ldots, p)$ and $t = t_n + vh = t_{pk+l} + vh \in (k\tau, (k+1)\tau]$, $v \in (0, 1]$. Obviously, there exists a constant $\tilde{R}$ such that

$$\left|\prod_{\mu=1}^{k+1} r(\mu\tau, (\mu - 1)\tau)\right| \leqslant \tilde{R}.$$

From the above analysis, we have the following inequality:

$$|e_h(t)| \leqslant R\tilde{R}\int_0^t |r(t, s)(q(s)e_h(s - \tau) + \delta_h(s))|ds, \tag{29}$$

where $\int_0^t |r(t, s)(q(s)e_h(s - \tau) + \delta_h(s))|ds$ can be expressed as

$$\int_0^t |r(t, s)(q(s)e_h(s - \tau) + \delta_h(s))|ds$$
$$= \sum_{i=0}^{n-1} h\int_0^1 |r(t, t_i + sh)(q(t_i + sh)e_h(t_i + sh - \tau) + \delta_h(t_i + sh))|ds$$
$$+ h\int_0^v |r(t, t_n + sh)(q(t_n + sh)e_h(t_n + sh - \tau) + \delta_h(t_n + sh))|ds$$
$$=: \sum_{i=0}^{n-1} h\int_0^1 \phi_n(t_i + sh)ds + h\int_0^v \phi_n(t_n + sh)ds.$$

Now, using an interpolatory m-point quadrature formula with collocation parameters $\{c_i\}$ to approximate $\int_0^1 \phi_n(t_i + sh)ds$, we have

$$\int_0^1 \phi_n(t_i + sh)ds = \sum_{j=1}^m b_j \phi_n(t_i + c_j h) + E_n^i(v) = E_n^i(v), \tag{30}$$

where $v \in (0,1](l < n)$ and $E_n^i$ indicates quadrature errors. So, we have

$$\int_0^t |r(t,s)(q(s)e_h(s - \tau) + \delta_h(s))|ds$$
$$= \sum_{i=0}^{n-1} h E_n^i(v) + h \int_0^v \phi_n(t_n + sh)ds. \tag{31}$$

By the orthogonality condition (22) and the Peano theorem, it is obvious that quadrature errors satisfy

$$\left| E_n^{(i)}(v) \right| \leqslant Q_i h^{m+1}, v \in [0,1], i \leqslant n - 1, \tag{32}$$

where $Q_i$ are constants. According to (29), (31) and (32), we can obtain

$$|e_h(t)|$$
$$\leq R\tilde{R} \sum_{i=0}^{n-1} h E_n^i(v) + R\tilde{R}h \int_0^v \phi_n(t_n + sh)ds$$
$$\leqslant R\tilde{R} \sum_{i=0}^{n-1} h E_n^i(v) + R\tilde{R}h \int_0^v |r(t, t_n + sh)\delta_h(t_n + sh)|ds$$
$$+ R\tilde{R}h \int_0^v |r(t, t_n + sh)q(t_n + sh)e_h(t_n + sh - \tau)|ds$$
$$\leqslant R\tilde{R} \sum_{i=0}^{n-1} h Q_i h^{m+1} + R\tilde{R}h r_0 \|\delta_h\|_\infty + R\tilde{R}h r_0 \tilde{r}_0 C_0 M_{m+1} h^m,$$

where $r_0 = \max_{t \in I} \int_0^t |r(t,s)|ds, \tilde{r}_0 = \max_{t \in I} |q(t)|$. By (27), we have

$$|e_h(t)| \leqslant R\tilde{R}Q \left( \sum_{i=0}^{n-1} h \right) h^{m+1} + R\tilde{R}r_0 D_1 M_{m+1} h^{m+1} + R\tilde{R}h r_0 \tilde{r}_0 C_0 M_{m+1} h^m$$
$$\leqslant \left( R\tilde{R}QT + R\tilde{R}r_0 D_1 M_{m+1} + R\tilde{R}r_0 \tilde{r}_0 C_0 M_{m+1} \right) h^{m+1}$$
$$=: C_2 h^{m+1}.$$

Here, $Q := \max\{Q_i : 0 \leqslant i \leqslant n - 1\}$. The estimation (24) follows from (26). The proof is completed. □

**Theorem 4.** *Assume that the solution of (1) lies in $C^{m+k}(I)(1 \leqslant k \leqslant m)$ and the m distinct collocation parameters $\{c_i\}$ are selected such that the general orthogonality condition (33) holds, with $J_k \neq 0$,*

$$J_v := \int_0^1 s^v \prod_{i=1}^m (s - c_i)ds = 0, v = 0, 1, .., k - 1. \tag{33}$$

*Then, for all meshes $I_h := \{t_0, t_1, \ldots\}$ with $h \in (0, \bar{h})$, the collocation solution $u_h$ with the above collocation parameters $\{c_i\}$ satisfies*

$$\max\{|y(t) - u_h(t)| : t \in I_h\} \leqslant C_4 h^{m+k}, \tag{34}$$

*where $C_4$ is a constant and independent of h.*

**Proof.** When $v = 0$, (31) is changed into

$$\int_0^t |r(t,s)(q(s)e_h(s-\tau) + \delta_h(s))|ds = \sum_{i=0}^{n-1} hE_n^i(0). \qquad (35)$$

Due to the general orthogonality condition (33) and the Peano theorem for quadrature, we can obtain

$$\left| E_n^{(i)}(0) \right| \leqslant Q_i h^{m+k}, i \leqslant n-1. \qquad (36)$$

Then, on meshes $I_h$, by (31), we have

$$
\begin{aligned}
|e_h(t)| &\leqslant R\tilde{R} \int_0^t |r(t,s)(q(s)e_h(s-\tau) + \delta_h(s))|ds \\
&= R\tilde{R} \sum_{i=0}^{n-1} hE_n^i(0) \leqslant R\tilde{R} \sum_{i=0}^{n-1} hQ_i h^{m+k} \\
&\leqslant R\tilde{R}Q\left(\sum_{i=0}^{n-1} h\right) h^{m+k} \leqslant R\tilde{R}QTh^{m+k} \\
&:= C_4 h^{m+k}.
\end{aligned}
\qquad (37)
$$

The proof is completed. □

### 5. Numerical Experiments

In the last section, two examples are given to illustrate the conclusions. Consider two IDDEs as follows:

$$
\begin{cases}
y'(t) = -2y(t) + y(t-1), & t \neq k, t \in I, \\
\triangle y = 0.2(-1)^k y, & t = k, \\
y(t) = 1, & t \in [-1, 0],
\end{cases}
\qquad (38)
$$

$$
\begin{cases}
y'(t) = -2ty(t) + ty(t-1), & t \neq k, t \in I, \\
\triangle y = -0.2y, & t = k, \\
y(t) = 1, & t \in [-1, 0].
\end{cases}
\qquad (39)
$$

In Figure 1, the image of the 2-Lobatto IIIA collocation solution with $p = 2$ for (38) is drawn. In Figure 2, we use the same method to draw the image for (39).

Tables 1 and 2 illustrate the ratios of the absolute errors between $p = 8$ and $p = 16$ at non-impulsive nodes and impulsive nodes using four different collocation methods for (38). Tables 3 and 4 illustrate the ratios of the absolute errors between $p = 8$ and $p = 16$ at non-impulsive nodes and impulsive nodes using four different collocation methods for (39). We can obtain that the convergence orders of the 2-Lobatto IIIA, 2-Radau IIA , 2-Gauss methods and 3-Gauss methods are $2, 3, 4$ and $6$, respectively. The ratios indicate that our numerical process can preserve the convergence order of collocation methods for IDDEs.

**Table 1.** The absolute error of 2-Lobatto IIIA and 2-Gauss methods for (38).

| $p$ | 2-Lobatto IIIA | | 2-Gauss | |
|---|---|---|---|---|
| | t = 0.5 | t = 1 | t = 0.5 | t = 1 |
| 2 | $1.7240 \times 10^{-2}$ | $1.2168 \times 10^{-2}$ | $2.7472 \times 10^{-4}$ | $2.0228 \times 10^{-4}$ |
| 4 | $3.9397 \times 10^{-3}$ | $2.8676 \times 10^{-3}$ | $1.4879 \times 10^{-5}$ | $1.0948 \times 10^{-5}$ |
| 8 | $9.3972 \times 10^{-4}$ | $7.0782 \times 10^{-4}$ | $1.0017 \times 10^{-6}$ | $7.3700 \times 10^{-7}$ |
| 16 | $2.3991 \times 10^{-4}$ | $1.7640 \times 10^{-4}$ | $6.2500 \times 10^{-8}$ | $4.6000 \times 10^{-8}$ |
| Ratio | 3.9170 | 4.0125 | 16.0272 | 16.0217 |

**Table 2.** The absolute error of 2-Radau IIA and 3-Gauss methods for (38).

| $p$ | 2-Radau IIA | | 3-Gauss | |
|---|---|---|---|---|
| | t = 0.5 | t = 1 | t = 0.5 | t = 1 |
| 2 | $2.1397 \times 10^{-3}$ | $1.5676 \times 10^{-3}$ | $1.8968 \times 10^{-6}$ | $1.3955 \times 10^{-6}$ |
| 4 | $2.3972 \times 10^{-4}$ | $1.6764 \times 10^{-4}$ | $2.8791 \times 10^{-8}$ | $2.1183 \times 10^{-8}$ |
| 8 | $3.7523 \times 10^{-5}$ | $2.7605 \times 10^{-5}$ | $4.4659 \times 10^{-10}$ | $3.2858 \times 10^{-10}$ |
| 16 | $4.8319 \times 10^{-6}$ | $3.5550 \times 10^{-6}$ | $6.9650 \times 10^{-12}$ | $5.1240 \times 10^{-12}$ |
| Ratio | 7.7657 | 7.7651 | 64.1195 | 64.1261 |

**Table 3.** The absolute error of 2-Lobatto IIIA and 2-Gauss methods for (39).

| $p$ | 2-Lobatto IIIA | | 2-Gauss | |
|---|---|---|---|---|
| | t = 0.5 | t = 1 | t = 0.5 | t = 1 |
| 2 | $1.0600 \times 10^{-2}$ | $1.6060 \times 10^{-2}$ | $1.6962 \times 10^{-4}$ | $2.6848 \times 10^{-4}$ |
| 4 | $2.7996 \times 10^{-3}$ | $3.8603 \times 10^{-3}$ | $1.0462 \times 10^{-5}$ | $1.5195 \times 10^{-5}$ |
| 8 | $6.9520 \times 10^{-4}$ | $9.6125 \times 10^{-4}$ | $6.5040 \times 10^{-7}$ | $9.2360 \times 10^{-7}$ |
| 16 | $1.7417 \times 10^{-4}$ | $2.3971 \times 10^{-4}$ | $4.0600 \times 10^{-8}$ | $5.7300 \times 10^{-8}$ |
| Ratio | 3.9915 | 4.0101 | 16.0197 | 16.1187 |

**Table 4.** The absolute error of 2-Radau IIA and 3-Gauss methods for (39).

| $p$ | 2-Radau IIA | | 3-Gauss | |
|---|---|---|---|---|
| | t = 0.5 | t = 1 | t = 0.5 | t = 1 |
| 2 | $1.4042 \times 10^{-3}$ | $1.5269 \times 10^{-3}$ | $3.1785 \times 10^{-7}$ | $4.0601 \times 10^{-6}$ |
| 4 | $1.9795 \times 10^{-4}$ | $2.1244 \times 10^{-4}$ | $8.6487 \times 10^{-9}$ | $6.6899 \times 10^{-8}$ |
| 8 | $2.5980 \times 10^{-5}$ | $2.8380 \times 10^{-5}$ | $1.4918 \times 10^{-10}$ | $1.0551 \times 10^{-9}$ |
| 16 | $3.3200 \times 10^{-6}$ | $3.6800 \times 10^{-6}$ | $2.3850 \times 10^{-12}$ | $1.6521 \times 10^{-11}$ |
| Ratio | 7.8253 | 7.7120 | 62.5489 | 63.8865 |

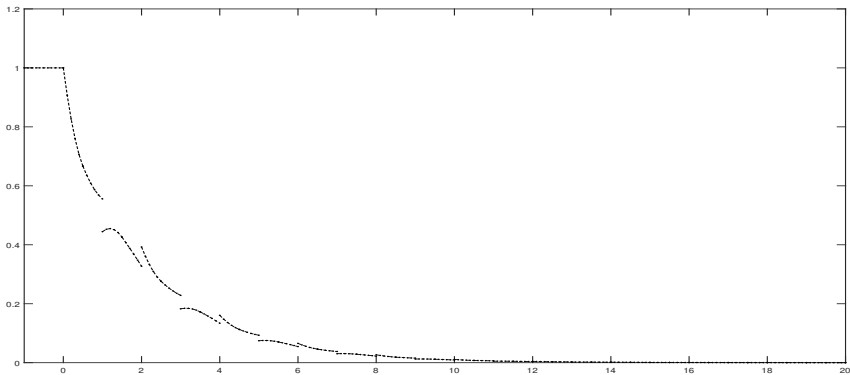

**Figure 1.** Two-stage Lobatto IIIA for (38).

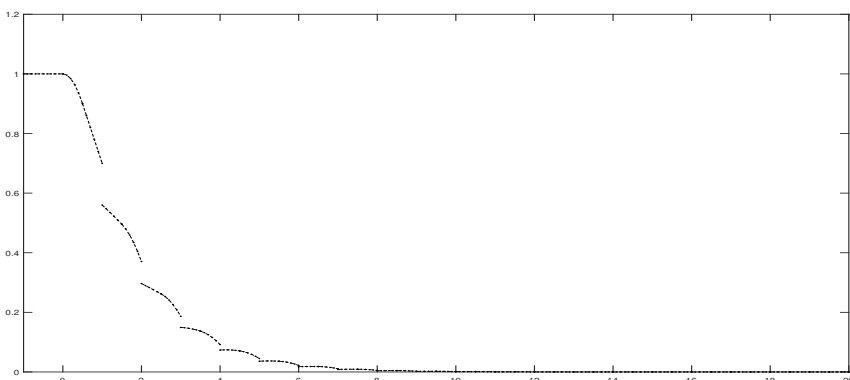

**Figure 2.** Two-stage Lobatto IIIA for (39).

**Author Contributions:** Conceptualization, Z.W.; Methodology, Z.W. Software, Z.W.; Validation, Z.W.; Formal analysis, Z.W.; Resources, G.Z.; Data curation, Z.W.; Writing—original draft, Z.W.; Writing—review&editing, Z.W. and G.Z.; Visualization, Z.W.; Supervision, G.Z.; Project administration, G.Z. and Y.S. All authors have read and agreed to the published version of the manuscript.

**Funding:** This research received no external funding.

**Data Availability Statement:** Not applicable.

**Conflicts of Interest:** The authors declare no conflicts of interest.

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
