# Peer review of "Convergence of Collocation Methods for One Class of Impulsive Delay Differential Equations"

_axioms, doi:10.3390/axioms12070700_

Round 1

Author Response

Report on ”Convergence of collocation methods for one class of impulsive delay differential equations” by Z.W. Wang, G.L. Zhang, Y. Sun

The authors investigate the collocation methods for the impulsive delay differential equation (1.1) (denoted in the manuscript). They study the global convergence, the global superconvergence and the local superconvergence of collocation solution of (1.1). The obtained results are interesting. There are some remarks presented below:

- Page 2, line 17+: define the elements tn for the interval (tn,tn+1) ,and the connection between tn and h; in addition, present all points used in the approximation of solutionfor problem (1.1);

- Page 3, line 12-: It is better authors to explain how Theorem 1 is obtained; eventually to write the results which are the base of the proof of Theorem 1;

- Page 3, line 4-: write the assumption for ∅;

- Page 4, line 7-: the formula for eh(t)  is written twice (see Page 4, line 3+);

- Page 8, line 11-: verify the number of ”Theorem 3.1”;

- To have a good connection between Sections 2,3,4, and Section 5, it is better to mention in the theoretical part of the paper something about 2-Lobatto IIIA, 2-RadauIIA, 2-Gauss, 3-Gauss collocation solutions; besides, in Page 11,reformulate the phrase from lines 3-,2-.

The paper must be revised according to the above remarks.

Dear Reviewer and Editor:

Greetings!

First of all, thank you very much for taking the time out of your busy schedule to read and revise my article. Thank you for your valuable suggestions. You have made comprehensive corrections to the structure, content, research methods, and results of my paper. It has played a very important role in improving the quality of my paper.

I have carefully studied the comments of the reviewers and made careful revisions to the paper based on the suggestions, as follows:

  1. Comment one: Page 2, line 17+: define the elements tn for the interval (tn,tn+1) , and the connection between tn and h; in addition, present all points used in the approximation of solution for problem (1.1);

         Answer one: The tn are fix time and tn=nh(n=0,1,…Np). The collocation        solution uh(t) is an approximation of the exact solution y(t) where t∈(tn,tn+1] . When t≠kτ (k=1,2,...,) , the right limit of uh(t) is uh(t).  When  t≠kτ (k=1,2,...,), the right limit of uh(t) is (1+Bk)uh(t).”  tn=nh are fix time” has been added to Page  2, line 17.

  1. Comment two: Page 3, line 12-: It is better authors to explain how Theorem 1 is obtained; eventually to write the results which are the base of the proof of Theorem 1.

Answer two: Theorem 1 can be obtained from the conclusion on page 8 of Butcher's book [H. Brunner, Collocation methods for Volterra integral and related functional differential equations, Cambridge university press, 2004].

  1. Comment three: Page 3, line 4-: write the assumption for ∅;

Answer three: The Minuscule ∅ in the fourth line on page 3 were wrongly written as capital letters ∅, which has been corrected. ∅ represents the initial function of the impulse delay differential equation. When t≤0,the collocation solution uh(t) is the same as the initial function ∅.

  1. Comment four: Page 4, line 7-: the formula for eh(t) is written twice (see Page 4, line 3+)

Answer four: The duplicate formula eh(t) has been deleted.

  1. Comment five: Page 8, line 11-: verify the number of ”Theorem 3.1”;

Answer five: Relevant proof has been provided as follows. “The (4.3) can be obtained by (3.13). The following discussion is for (4.2).” has been added to Page8.

  1. Comment six: To have a good connection between Sections 2,3,4, and Section 5, it is better to mention in the theoretical part of the paper something about 2-Lobatto IIIA, 2-RadauIIA, 2-Gauss, 3-Gauss collocation solutions; besides, in Page 11, reformulate the phrase from lines 3-,2-.

Answer six: 6. All these 2-Lobatto IIIA, 2-RadauIIA, 2-Gauss, 3-Gauss collocation methods satisfy the convergence properties proposed in this paper. “We choose v = 0 on meshes  Ih, then (4.10) can be expressed as” in page 11 is changed into “When v=0, (4.10) is changed into”.

Modifications:

  1. ”tn=nh are fix time” has been added to Page 2, line 17.
  2. Explained how theorem 1 was obtained.
  3. The correct letter ∅ has been written on Page 3, line 4.
  4. The duplicate formula eh(t) on Page 4, line 7 has been deleted.
  5. Relevant proof has been provided as follows. “The (4.3) can be obtained by (3.13). The following discussion is for (4.2).” has been added to Page8.
  6. “We choose v = 0 on meshes Ih , then (4.10) can be expressed as” in page 11 is changed into “When v=0, (4.10) is changed into”.

Reviewer 2 Report

 Minor editing of English language required

Author Response

Convergence of collocation methods for one class of impulsive delay differential equations

In the work Convergence of collocation methods for one class of impulsive delay differential

equations, the authors considered a collocation method for a class of impulsive delay differential equations. The authors discussed convergence, global superconvergence, and local superconvergence. It is not clear from the paper which results are known and which are not. Many known results are not cited.

Comments:

Authors must be careful about plagiarism. Eq. (1.1) is not their invention. The paper must be

written carefully and the differences at least compared to the following works should be taken

into account:

-D.D. Bainov, M.B. Dimitrova, A.B. Dishliev, Oscillation of the solutions of a class of impulsive

differential equations with deviating argument, J. Appl. Math. Stoch. Anal. 11 (1) (1998) 95-102.;

-J. Yan, C. Kou, Oscillation of solutions of impulsive delay differential equations, J. Math. Anal. Appl. 254 (2) (2001) 358-370;

-Ravi P. Agarwal and Fatma Karakoc, A survey on oscillation of impulsive delay differential equations, Computers and Mathematics with Applications 60 (2010) 1648-1685.

Additional comments:

-English must be revised. There are meaningless sentences (page 1, line 16...);

-page 1, line 25: give examples of works;

-page 2, line 45, either polynomial or series, it is not clear;

-page 2, Eq. (2.1), after "[...]" follows "," and not ".". Correct the punctuation throughout the work;

-page 2, The "Collocation methods" section is written without citations. Complete the entire paper with citations where appropriate.

Dear Reviewer and Editor:

Greetings!

First of all, thank you very much for taking the time out of your busy schedule to read and revise my article. Thank you for your valuable suggestions. You have made comprehensive corrections to the structure, content, research methods, and results of my paper. It has played a very important role in improving the quality of my paper.

I have carefully studied the comments of the reviewers and made careful revisions to the paper based on the suggestions, as follows:

  1. Comment one: Authors must be careful about plagiarism. Eq. (1.1) is not their invention. The paper must be written carefully and the differences at least compared to the following works should be taken into account:

-D.D. Bainov, M.B. Dimitrova, A.B. Dishliev, Oscillation of the solutions of a class of impulsive differential equations with deviating argument, J. Appl. Math. Stoch. Anal. 11 (1) (1998) 95-102.;

-J. Yan, C. Kou, Oscillation of solutions of impulsive delay differential equations, J. Math. Anal. Appl. 254 (2) (2001) 358-370;

-Ravi P. Agarwal and Fatma Karakoc, A survey on oscillation of impulsive delay differential equations, Computers and Mathematics with Applications 60 (2010) 1648-1685.

Answer one: To our knowledge, the oscillation of this impulse delay differential equation has been studied in these three articles provided by the reviewers, but the use of collocation methods to solve this equation has not been studied yet. The main focus of this article is on the convergence research of collocation methods rather than oscillation. The focus of this article is on numerical solutions rather than exact solutions. The relevant literature on this type of equation has been supplemented in the article.

  1. Comment two: English must be revised. There are meaningless sentences (page 1, line 16...);

Answer two: English has been revised. The meaningless sentence on page 1, line 16 has been deleted.

  1. Comment three: page 1, line 25: give examples of works;

Answer three: Because to the best of our knowledge, there are no articles referring to the convergence of the collocation method for IDDEs. The original English has been revised.   “Because to the best of our knowledge, there are few articles referring to the convergence of the collocation method for IDDEs.” in page 1 is changed into “Because to the best of our knowledge, there are no articles referring to the convergence of the collocation method for IDDEs.”.

  1. Comment four: page 2, line 45, either polynomial or series, it is not clear;

Answer four: This article first provides a common definition that denotes the space of all real polynomials of degree not exceeding m. The purpose of defining this space is to select the collocation solution in the space. The collocation solution should be a polynomial within (tn,tn+1] which degree does not exceed m. The specific collocation solution not only exists in the space, but also satisfies the equation (2.1) on page 2.

  1. Comment five: page 2, Eq. (2.1), after "[...]" follows "," and not ".". Correct the punctuation throughout the work;

Answer five: The punctuation has been corrected.

  1. Comment six: page 2, The "Collocation methods" section is written without citations. Complete the entire paper with citations where appropriate.

      Answer six: The relevant references have been supplemented on page 2, line 43 and line 47. [H. Brunner, Collocation methods for Volterra integral and related functional differential equations, Cambridge university press, 2004.], [H. Liang and Hermann Brunner, Collocation methods for differential equations with piecewise linear delays, Communications on Pure and Applied Analysis, 11 (2012), 1839-1857.].

Modifications:

  1. The relevant literature on this type of equation has been supplemented in the article on page 1 line 14. [D.D. Bainov, M.B. Dimitrova, A.B. Dishliev, Oscillation of the solutions of a class of impulsive differential equations with deviating argument, J. Appl. Math. Stoch. Anal. 11 (1) (1998) 95-102.], [J. Yan, C. Kou, Oscillation of solutions of impulsive delay differential equations, J. Math. Anal. Appl. 254 (2) (2001) 358-370],[Ravi P. Agarwal and Fatma Karakoc, A survey on oscillation of impulsive delay differential equations, Computers and Mathematics with Applications 60 (2010) 1648-1685.].
  2. The meaningless sentence on page 1, line 16 has been deleted.
  3. “Because to the best of our knowledge, there are few articles referring to the convergence of the collocation method for IDDEs.” in page 1 is changed into “Because to the best of our knowledge, there are no articles referring to the convergence of the collocation method for IDDEs.”.
  4. Provide a more detailed description of the meaning of polynomials or series in this revised letter.
  5. The punctuation has been corrected.
  6. The relevant references have been supplemented on page 2, line 43 and line 47. [H. Brunner, Collocation methods for Volterra integral and related functional differential equations, Cambridge university press, 2004.], [H. Liang and Hermann Brunner, Collocation methods for differential equations with piecewise linear delays, Communications on Pure and Applied Analysis, 11 (2012), 1839-1857.].

Round 2

Reviewer 2 Report

The authors took into account my observations. There are still typos ("," after 3.2, R_{m+1}, 3.9, ...).
Since after acceptance the authors can no longer modify the text, my advice is to carefully reread the text to correct any grammatical mistakes (The Ratios->The ratios 9.2, ...).
Another iteration is not necessary. The work can be accepted from my point of view.

Author Response

The authors took into account my observations. There are still typos ("," after 3.2, R_{m+1}, 3.9, ...).

Since after acceptance the authors can no longer modify the text, my advice is to carefully reread the text to correct any grammatical mistakes (The Ratios->The ratios 9.2, ...).

Dear Reviewer and Editor:

Greetings!

We sincerely thank the reviewer for careful reading. Thanks for your careful checks. We were really sorry for our careless mistakes. Thank you for your reminder. In our resubmitted manuscript, the typo is revised. Thanks for your correction. As suggested by the reviewer, we have added “,” on page 3 (3.2); page 4 R_{m+1}; page 5 (3.9); page 7 line 12; page 7 line 14. We also have changed the “,” into “.” on page 6 line 20.

We also have changed the “The Ratios” into “The ratios” on page 11; the “satisfy” into “satisfies” on page 8 line 7.
